# Fully automated fast-flow synthesis of antisense phosphorodiamidate morpholino oligomers

Chengxi Li[1,9], Alex J. Callahan[1,9], Mark D. Simon[1], Kyle A. Totaro[1], Alexander J. Mijalis[1], Kruttika-Suhas Phadke[2], Genwei Zhang[1], Nina Hartrampf [1,8], Carly K. Schissel[1], Ming Zhou[3], Hong Zong[3], Gunnar J. Hanson[3], Andrei Loas[1], Nicola L. B. Pohl [4], David E. Verhoeven[2] & Bradley L. Pentelute [1,5,6,7 ✉]

Rapid development of antisense therapies can enable on-demand responses to new viral pathogens and make personalized medicine for genetic diseases practical. Antisense phosphorodiamidate morpholino oligomers (PMOs) are promising candidates to fill such a role, but their challenging synthesis limits their widespread application. To rapidly prototype potential PMO drug candidates, we report a fully automated flow-based oligonucleotide synthesizer. Our optimized synthesis platform reduces coupling times by up to 22-fold compared to previously reported methods. We demonstrate the power of our automated technology with the synthesis of milligram quantities of three candidate therapeutic PMO sequences for an unserved class of Duchenne muscular dystrophy (DMD). To further test our platform, we synthesize a PMO that targets the genomic mRNA of SARS-CoV-2 and demonstrate its antiviral effects. This platform could find broad application not only in designing new SARS-CoV-2 and DMD antisense therapeutics, but also for rapid development of PMO candidates to treat new and emerging diseases.

---

[1] Department of Chemistry, Massachusetts Institute of Technology, Cambridge, MA, USA. [2] Department of Veterinary Microbiology and Preventive Medicine, College of Veterinary Medicine, Iowa State University, Ames, IA, USA. [3] Sarepta Therapeutics, Cambridge, MA, USA. [4] Department of Chemistry, Indiana University, Bloomington, IN, USA. [5] The Koch Institute for Integrative Cancer Research, Massachusetts Institute of Technology, Cambridge, MA, USA. [6] Center for Environmental Health Sciences, Massachusetts Institute of Technology, Cambridge, MA, USA. [7] Broad Institute of MIT and Harvard, Cambridge, MA, USA. [8] Present address: University of Zurich, Department of Chemistry, Zurich, Switzerland. [9] These authors contributed equally: Chengxi Li, Alex J. Callahan. ✉email: blp@mit.edu

New and emerging pathogens, such as COVID-19 necessitate rapid drug development, and typical drug development pipelines are ill-suited to meet these demands. In contrast, the logical and rapid development of antisense drugs make them well-suited to this area[1–3]. The strength of such a strategy is highlighted by the emergence of SARS-CoV-2, wherein oligonucleotide-based drugs were among the first treatment types to enter human trials as early as 2 months after the virus's first reports[4–6]. Although these results are promising, rapid preclinical development of antisense drugs necessitates new technologies to enable high-throughput drug screening.

Antisense therapeutics are synthetic RNA mimetics that bind to mRNA via complementary base-pairing interactions to modulate transcription[1]. Because their design is informational in nature[7], they can target virtually any genetic disease in a rapid and logical fashion. Among antisense compound types, phosphorodiamidate morpholino oligomers[8–10] (PMOs) are the most tested as antiviral agents, with clinical trials already underway for treatments against Dengue[11], Marburg[12], Ebola[12], Influenza[13], West Nile[14], and SARS-CoV-1 (refs. [15,16]) viruses. PMO backbones are derived from RNA, wherein the five-membered ribosyl ring has been replaced with a six-membered morpholino ring, and phosphate linkages have been replaced with uncharged phosphorodiamidates (Fig. 1a). These modifications make PMOs resistant to nucleases[9], while retaining strong binding affinity for target RNA. Although PMOs are known to be effective antisense candidates, robust screening efforts for efficient transition to the clinic have remained elusive. Tedious synthesis protocols limit the production of screening libraries needed for sequence optimization, and development timelines remain long.

Changes to the chemistry of PMO synthesis are greatly needed to enable the rapid drug development. PMOs are synthesized from the 5′- to the 3′-end on a crosslinked polystyrene solid support[8,17–19] (Fig. 1b), with coupling times on the order of 180 min (refs. [17,20]). With therapeutic PMO sequences on the order of 20 residues, synthesis times are on the order of weeks. Unsurprisingly, the production of screening libraries places a significant burden on the development of PMO drugs (Fig. 1c) due to lengthy protocols. To address this limitation, it is common to run synthesis reactions on automated systems[17]. In theory,

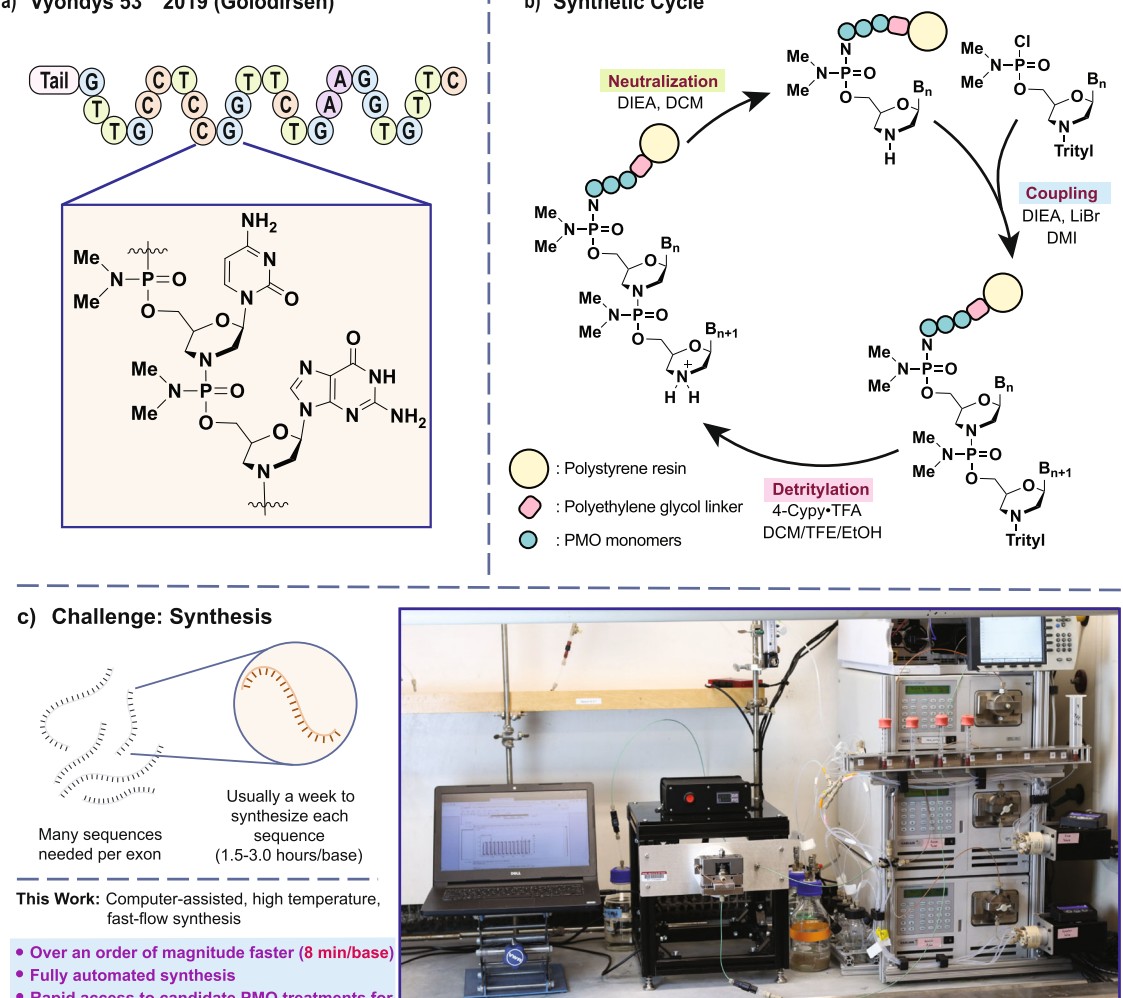

**Fig. 1 Automated fast-flow technology can produce on-demand customized PMO sequences. a** The structure of PMO drug Golodirsen is shown. **b** Each nucleotide is incorporated as a trityl-protected phosphoramido chloridate in 1,3-dimethyl-2-imidazolidinone (DMI). Detritylation frees the 3′-terminal amine with the conjugate acid of a non-nucleophilic heterocycle, typically 4-cyanopyridine in dichloromethane–trifluoroethanol (DCM/TFE) mixtures. Neutralization of the 3′-amine prepares the chain for the next coupling reaction using tertiary amines in DCM/isopropanol mixtures. **c** Development of PMO drugs requires testing of many sequences and laborious production of sequential revisions. An automated fast-flow synthesizer allows for rapid manufacture of PMO sequences at 90 °C.

such instruments that combine automation with flow chemistry can mix and heat reagents with efficiencies that are unattainable by batch methods[21–23], but application of these potential advantages to PMO synthesis has remaied elusive. Further work is needed to improve access to PMO compounds.

Rapid production of PMOs will be required to expand their application to new disease types. With the emergence of SARS-CoV-2 in late 2019, new techniques to develop drugs on a rapid timescale are greatly needed, and PMOs could find great application here. Accordingly, PMOs have been reported to treat SARS-CoV-1 (refs. [15,16]), but progress at the time was too slow to justify further development. Use of the same strategy to inhibit SARS-CoV-2 (Fig. 2a) should be possible in principle, but long development timelines have left this modality underexplored. Even for well-characterized strategies where PMOs have been successful, development has been slow. PMO treatments for Duchenne muscular dystrophy (DMD) are well-understood (Fig. 2b)[24–26] and applicable to a majority of DMD subtypes[27,28], but commercial treatments are available to only a small fraction of these.

In this work, we disclose an automated instrument that can expedite PMO synthesis by over an order of magnitude using chemistry done in flow. Enhanced reaction rates on the reported flow synthesizer result in nucleobase coupling in only 8 min. With this technique, we produce candidate PMOs for diseases with no current treatment in a matter of hours with our automated synthesis platform. A PMO designed to bind to the transcription regulatory stem loop of the SARS-CoV-2 genomic mRNA to inhibit the production of viral proteins is synthesized in only 3.5 h. This antiviral compound demonstrates highly specific inhibition of the native SARS-CoV-2 virus in live cells. Further optimization of this compound could provide a greatly needed stop-gap therapeutic for the most severe cases of COVID-19. The optimized synthesis protocol can also accelerate the drug development timelines for established applications of PMOs. We synthesize three therapeutic candidates for DMD in a single day. The automation technology demonstrated here can open the door to enable rapid screening of PMO drug candidates for all possible DMD subtypes.

## Results

**Design of an automated microscale flow synthesizer**. A microscale instrument was designed to carry out the flow synthesis of PMOs. The instrument was constructed from commercially available components and a machined reaction vessel, using a design similar to a previously reported fast-flow peptide synthesizer[29,30]. The base design consists of six modules connected in series (Fig. 3a). The first module is a collection of glass containers with liquid reagents stored under nitrogen. Two chemically inert valves compose the second module. Under computer control, each valve chooses its input from the available reagents in module 1, updating throughout a run. The third module is composed of two HPLC pumps, each connected to one of the outputs from module 2. Each pump is capable of supplying up to 2.5 mL per minute limited by reagent viscosity and pump configuration. The output streams from module 3 meet in a T-mixer and then travel to module 4, the reaction vessel module. Flow enters module 4 and is passed through a 90 cm long metal tube over a heated aluminum core bringing the solution to temperature in ~2 s. Module 4 holds the solid-phase resin in a removable reactor chamber 1 mL in volume at the desired temperature. The preheated flow passes through the resin, reacting with the growing PMO chains. Module 5 is a UV–vis detector used to monitor the composition of the spent reagent solution in-line. Module 6 is a computer that controls all other modules,

using a modular script in the Mechwolf programming environment[31].

This design allows for precise control of reaction conditions for microscale PMO synthesis. The reactor body is designed for a 4.4 μmol-scale synthesis, the equivalent of 10 mg of a medium loading resin (0.39–0.43 mmol/g). Reagent delivery is encoded using pump strokes and flow rate. Each stroke of the HPLC pump carries 40 μL of solvent and it takes 12 strokes from both pumps (960 μL) to reach the resin. Flow rate is adjusted by increasing both the time to deliver strokes, and the time between strokes. Reaction time on the resin can be increased by increasing the number of pump strokes, or by decreasing the flow rate once the reagents hit the resin. Clearance of regents from the reservoir takes 20 pump strokes (1.6 mL), so reaction steps are separated by washes of at least 20 strokes of the appropriate wash solvent.

The optimized synthesis sequence with controls for each module can be seen in Fig. 3b. A discussion about optimization of individual parts of the instrument is provided in the Supplementary section 10.

**Optimization of automated fast-flow synthesis**. Iterative changes in flow synthesis variables were used to develop a flow recipe that can produce PMO sequences of similar purity during room temperature syntheses. Solid-phase PMO synthesis is sensitive to small variations in reaction efficiency, as the many reactions in series amplify off-target pathways. We took advantage of this amplification to optimize reactions in flow. For each variable of interest, we synthesized a 4-mer PMO, a process that involves 12 sequential reaction steps. Using LC–MS, we compared the crude purity of the products from each reaction sequence. Timings and reagents for each synthesis were as shown in Fig. 3b with the modifications listed in Fig. 4a. The actions that the Python script sends to modules 2 and 3 throughout the synthesis cycle are shown in Fig. 3b. The resulting resin-bound PMO product was cleaved from the solid support, the sample analyzed by LC–MS, and the relative levels of the product and high molecular weight side products were quantified, using a molecular feature extraction utility. Of special interest were side products arising from incomplete couplings, and we tracked their relative abundance separately.

Use of the highest allowable temperature determined from monomer and resin-bound PMO stability studies, 90 °C, provided the cleanest crude PMO. Increasing the temperature of PMO synthesis significantly decreases reaction times, contingent on the stability of reagents. Temperatures of over 70 °C will accelerate both on-target and off-target reactions, with degradation of synthetic intermediates limiting the maximum possible synthesis temperature. To determine the maximum reaction temperature, we tracked the degradation of all synthetic components at a range of temperatures, and modified chemical variables to ensure stability of all components (see Supplemental Information sections 8 and 9). Initial reaction screens were carried out at 70 °C, a milder temperature that enables use of the standard 4-cyanopyridine trifluoroacetate deprotection solution without the significant degradation found at 90 °C (see Supplementary Fig. 11). Initial results from flow synthesis at 70 °C provided the desired material, but with a crude purity of 72%, lower than the benchmark 95% from room temperature syntheses (Fig. 4a, entries 1 vs 2). Further optimization with changes to the instrument command recipe, monomer equivalents, and coupling catalysts[20] improved crude purity to 92% (Fig. 4a, entries 3–6). Detectable levels of side products remained, so the improvement of the coupling reaction was still required. Although increasing monomer excess would likely increase crude purity, we capped this value at 18 equivalents, a generally accepted upper limit in

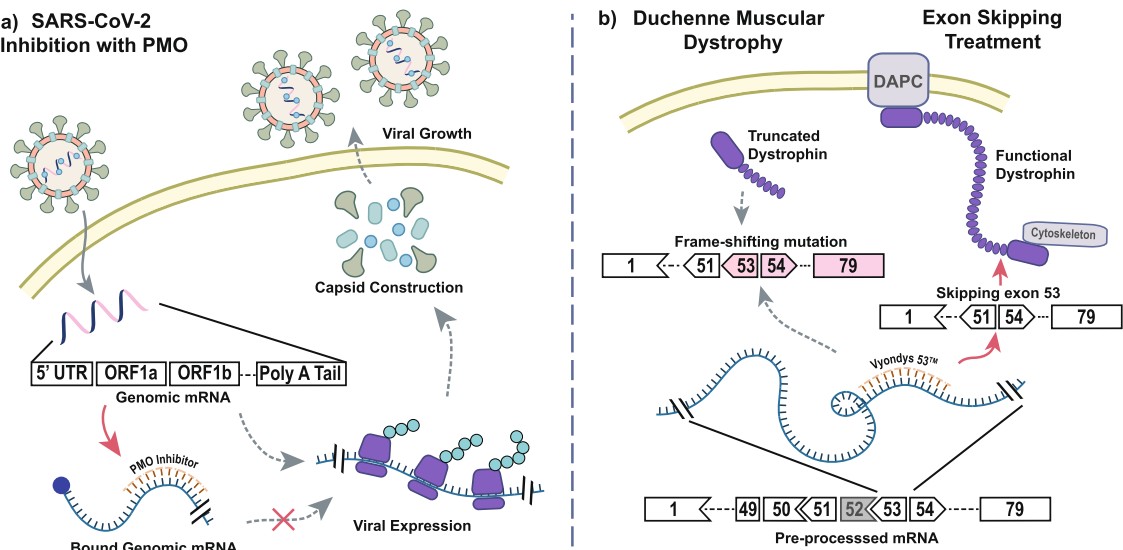

**Fig. 2 Antisense PMO therapeutics are adaptable to a range of disease types. a** An antisense PMO binding to the 5′ UTR of the SARS-CoV-2 genomic transcript can prevent expression of viral genes and subsequently inhibit viral growth. **b** Dystrophin is an integral membrane protein that anchors the cytoskeleton to the muscle cell membrane via the dystrophin associated protein complex (DAPC). Little to no natural dystrophin is produced in patients with Duchenne muscular dystrophy (DMD). The PMO Vyondys 53™ induces skipping of exon 53 to regain the proper reading frame, producing shorter but functional dystrophin.

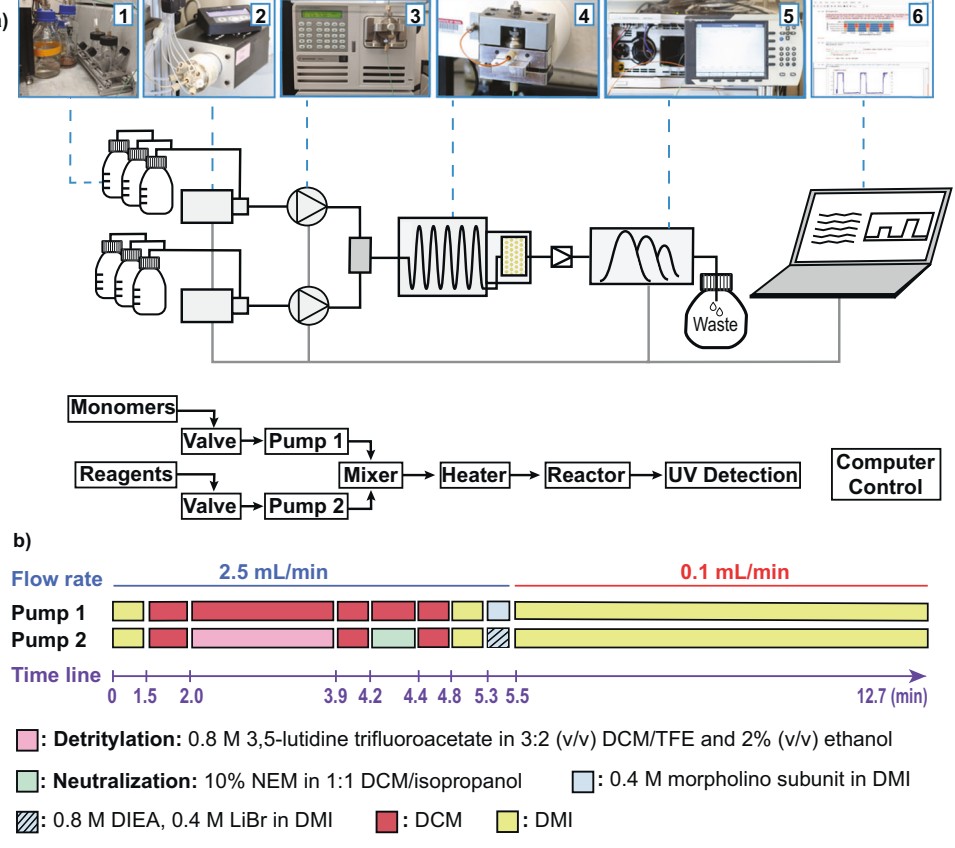

**Fig. 3 The automated fast-flow instrument features six modules that act to effect rapid PMO synthesis. a** A schematic for the flow path of the instrument is shown. Valves select the appropriate solutions from glass bottles under nitrogen pressure. Two HPLC pumps mix the reagents and flow them over a packed resin bed held at 90 °C. Effluent cools as it exits the reactor and passes through a back-pressure regulator to an in-line UV–vis detector, where reaction progress is monitored. A computer running an automated Python script controls the valves and pumps throughout the synthesis, and tracks the instrument performance from the UV–vis detector. **b** The optimized protocol for PMO synthesis is shown. Instructions are delivered by the control program to the pumps and valves at each of the listed times. Bars represent activity of the two HPLC pumps, and colors indicate which reagents the valves are open to. Reagent stocks are prepared in double the concentration that is intended to hit the resin. Dilution from the second pump prepares the correct concentrations upon mixing.

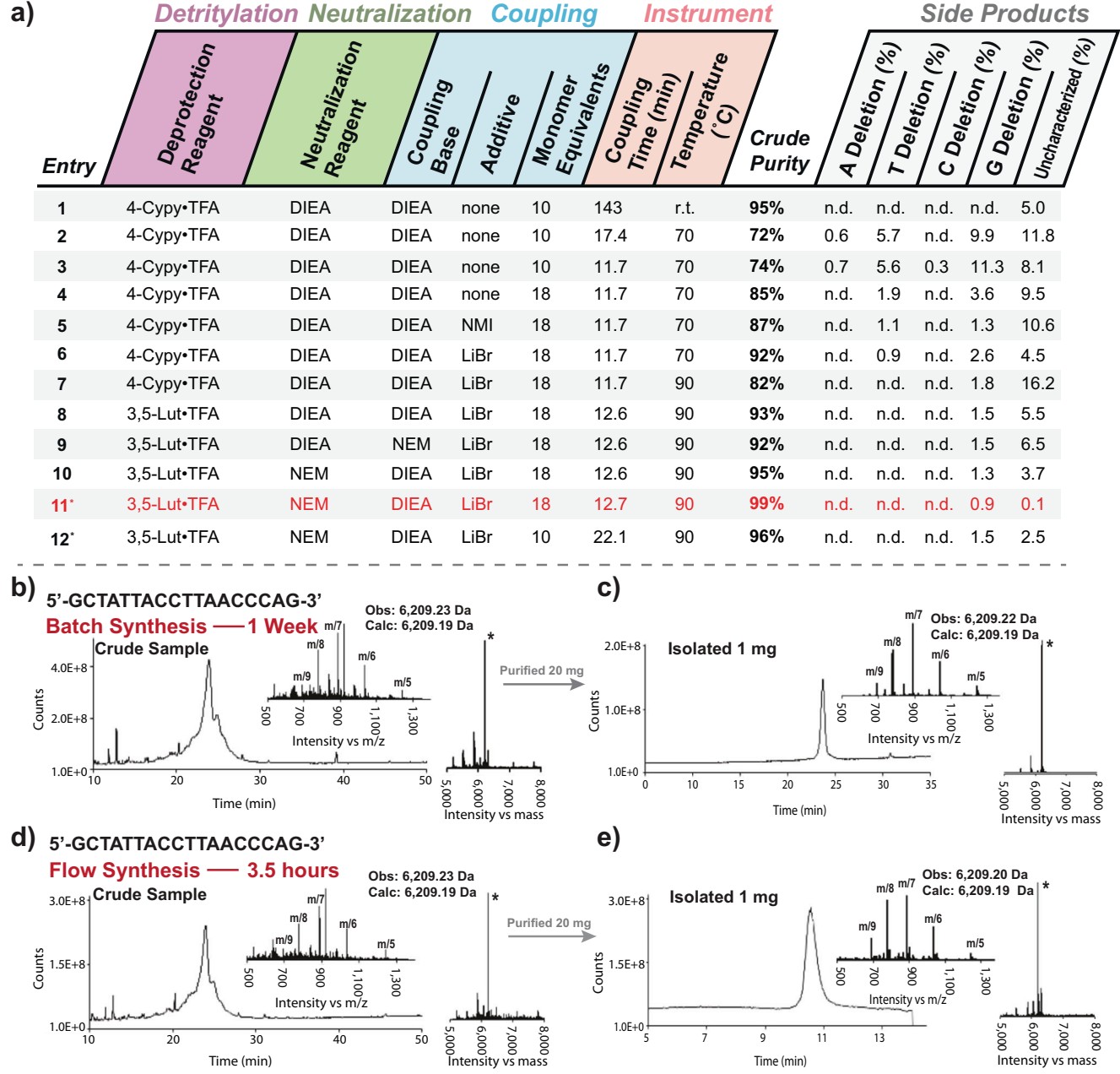

**Fig. 4 Optimized flow synthesis produces PMOs with similar purity to manual batch protocols. a** Synthesis variables were changed in sequence, and relative crude purities were tracked by comparison of the crude LC–MS traces. Crude purity and deletion sequences quantified, as described in Supplementary Section 6. *The Python script was edited to slowly deliver the monomers, reducing flow inaccuracies due to the viscous solution of the G monomer. N.d. = levels were below the sensitivity of detection with the described quantification method—not detected. Column colors indicate which category the variables correspond to pink—detritylation, green—neutralization, blue—coupling, and orange—instrument. Red lettering in-line 11 denotes the best observed conditions. **b** The total ion current chromatogram (TICC) of a batch-synthesized sample of the 18-mer PMO IVS2-654 is shown along with the mass spectrum and associated deconvoluted mass spectrum. **c** The TICC of the batch-synthesized sample after purification by cation exchange chromatography is shown along with the mass spectrum and associated deconvoluted mass spectrum. **d** The TICC, mass spectrum and deconvoluted mass spectrum are shown for the crude product from a flow synthesis of the same 18-mer sequence. **e** The TICC of the flow-synthesized sample after purification along with the mass spectrum and deconvoluted mass spectrum. Note: traces **c** and **e** were acquired using different LC–MS methods.

academic and patent literature[17,20]. We instead increased the temperature to 90 °C to improve coupling rates. Along with optimization of deprotection conditions, neutralization, and coupling bases (Fig. 4a, entries 7–10), the increase in temperature provided an optimized recipe that yields a crude purity of 99% (Fig. 4a, entry 11). This protocol was used for the production of the sequences reported in the remainder of this work.

The final experiment shown in Fig. 4a (entry 12) demonstrates that high temperature flow synthesis of PMOs does not require more monomer equivalents than room temperature protocols. The phosphoramido chloridate monomers are costly, and it is common to minimize the excess used. Using only ten equivalents of monomer at high temperature is effective, but requires a longer coupling step with this hardware configuration.

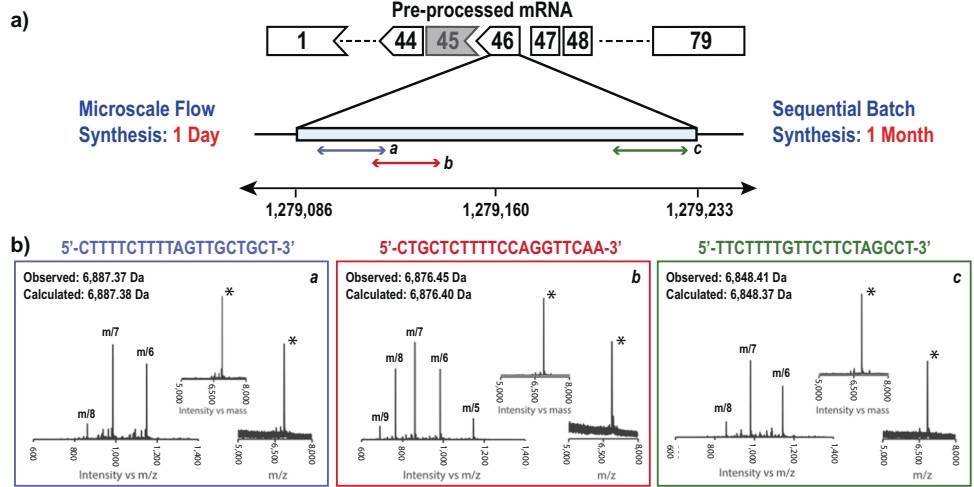

**Fig. 5 Automated flow synthesis enables the rapid production of antisense PMO candidates to treat DMD. a** Gene diagram showing the splicing sites where the three potential therapeutic sequences are targeted. Three sequences were chosen for targeting splice donor and acceptor sites of exon 46. **b** The mass spectrum, deconvoluted mass spectrum, and MALDI mass spectra of purified PMOs synthesized in flow are shown (see Supplementary section 13.4).

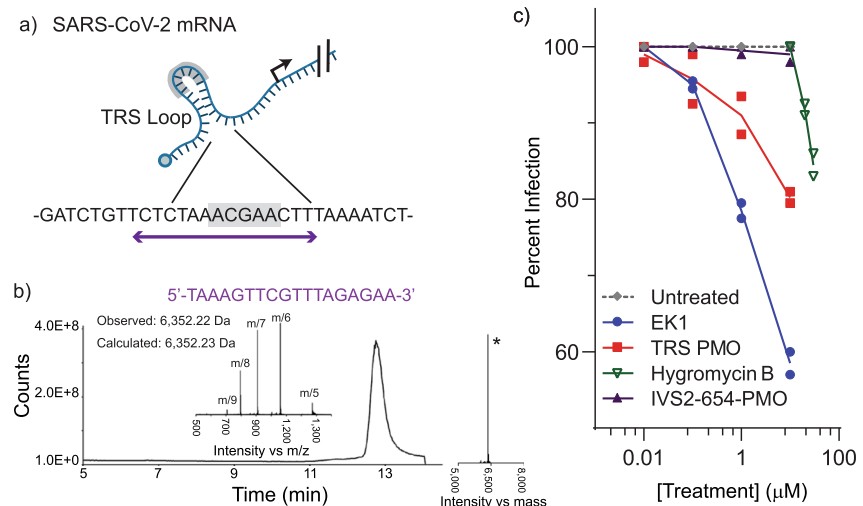

**Fig. 6 Synthetic PMO can be engineered to inhibit SARS-CoV-2 replication. a** The 5′ UTR TRS is conserved between coronaviruses, and is a target for antisense knockdown of viral replication. The TRS is shown in gray, and a sequence was chosen to encompass this region and surrounding bases, an effective strategy for targeting the SARS-CoV-1 mRNA. **b** The TICC, mass spectrum, and deconvoluted mass spectrum of the purified PMO sequence are shown. **c** Inhibition of native SARS-COV-2 by various treatments is shown. Percent inhibition represents observed viral reductions in culture supernatants as measured by qRT-PCR, as compared to untreated control. Increasing concentrations of TRS-PMO results in inhibition of viral replication. Inhibition of viral growth is observed, as previously reported for EK1 and hygromycin B. Measurement at each concentration is from $n = 2$ biological replicates, with each point averaging two technical replicates.

Application of the optimized protocol enabled the rapid synthesis of an 18-mer PMO, which hybridizes to the β-thalassemia gene sequence, IVS2-654 (refs. [32,33]). Using batch protocols, production of the PMO took one full week. The flow protocol enabled the production of the full PMO sequence in only 3.5 h, and the crude products were of comparable purity by LC–MS (Fig. 4b, d). From each synthesis 1 mg of PMO with >85% purity were obtained (10.1, 10.1, and 10.2% yields with respect to resin loading, 10 mg of 0.39–0.43 mmol/g; Fig. 4c, e).

**The microscale flow synthesizer rapidly produces potential DMD treatments**. We leveraged the platform for rapid production of three PMO sequences targeted to skip exon 46 of DMD pre-mRNA. Although exon skipping at this site has an important place in the history of splice alteration for DMD[34], no treatment

options are available that target exon 46. Three sequences near splice acceptor and donor sites were chosen, as previously reported[35] (Fig. 5a). Using the automated fast-flow instrument, the three 20-mer sequences were synthesized in a single day in succession. In each case, after cleavage and purification, 1 mg of PMO material was isolated with >85% purity. Manufacturing these three sequences under batch conditions would take ~1 month if done sequentially (Fig. 5b).

**Synthetic anti-SARS-CoV-2 PMO shows viral inhibition**. PMOs used as steric block antisense compounds provide a potential route to inhibit the viral replication. We designed a PMO sequence to inhibit SARS-CoV-2 replication and synthesized it in 3.5 h. Using room temperature synthesis, manufacturing each new potential COVID-19 therapeutic would

typically require at least a week. The 18-mer PMO was targeted to the transcription regulatory sequence (TRS) stem loop of the 5′ UTR of the genomic mRNA (Fig. 6a), a strategy that proved effective for treatment of the SARS-CoV-2 (ref. [36]) and closely related SARS-CoV-1 (refs. [15,16]). Synthesis and subsequent purification of one third of the crude sample yielded 0.5 mg of PMO with >90% purity (extrapolated yield of 5.5% with respect to resin loading, 10 mg of 0.39–0.43 mmol/g; Fig. 6b).

To test the antiviral activity of the synthesized PMO, we measured viral RNA levels after viral incubation with a low multiplicity of infection (MOI). Lower levels of viral RNA is consistent with slowed viral growth, and successful inhibition of viral protein production should lower the observed viral RNA. Prior to inoculation, Vero-E6 cells were incubated with either the TRS-PMO, IVS2-654-PMO as a negative control having no intracellular target present, or known inhibitors of viral growth as positive controls, hygromycin B[15] or EK1 (refs. [37,38]), a pan-coronavirus spike protein peptide-based fusion inhibitor. Cells were inoculated at a MOI of 0.1 for 2 h and grown with their respective treatments. Subsequent viral RNA levels were measured 72 h after infection. We observed dose-dependent reduction of viral RNA copies with increasing concentrations of the TRS-PMO, but observed no such inhibition with the same concentration of the IVS2-654-PMO (Fig. 6c). These results demonstrate specific inhibition of the SARS-CoV-2 virus with the synthetic PMO produced by our automated flow technology.

## Discussion

Automated fast-flow synthesis is a potentially valuable tool that capitalizes on the recent successes of PMO antisense treatments[24–26] to expand the potential of PMOs to treat new diseases. With rapid flow synthesis, the production of novel PMO therapeutics will not be burdened by long lead optimization cycles. The fully automated fast-flow synthesizer developed here enables rapid preheating of reagents, and efficient heat and mass transfer within the resin bed on a scalable, mechanically robust platform. With these features, we adapted PMO synthesis to 90 °C and ultimately succeeded in decreasing coupling time from 3 h per nucleotide to 8 min.

To demonstrate the power of this flow platform, we synthesized three candidates for a new DMD treatment in a single day, and we anticipate that the development of new PMO drugs using this platform will be similarly accelerated. Given the urgent circumstances around the global COVID-19 pandemic, such timeline reductions are needed, and we demonstrated the utility of fast PMO synthesis by producing a potential antisense antiviral treatment in only 3.5 h. The PMO therapeutic candidate described here demonstrates specific inhibition of the native SARS-CoV-2 virus. Given the favorable toxicological profile of PMO drugs[8], further improvements in efficacy could result in an effective treatment for SARS-CoV-2.

Overall, our results illustrate that machine control of flow chemistry can improve synthetic outcomes beyond what is possible with manual techniques. The strategy in this work is applicable to diverse polymer backbones, and we envision high temperature automated flow synthesis will enable the development of new on-demand biopolymers that may currently be inaccessible due to tedious, difficult, or impractical syntheses.

## Methods

**Flow PMO synthesis**. The following procedure was used for automated flow synthesis. Aminomethyl polystyrene resin loaded with a PEG₃ tail (see Supplementary section 3.1; 10 mg, 0.39–0.43 mmol/g loading) was loaded into the reactor, the reactor was connected to the reactor head and heated to 90 °C. DCM was delivered at 5 mL/min (2.5 mL/min per pump) for 30 s to remove air. The flow was stopped and the resin was allowed to swell at 90 °C for 5 min. The flow protocol was started with an initial DMI wash at 5 mL/min (2.5 mL/min per pump) for 90 s,

then a DCM wash at 5 mL/min (2.5 mL/min per pump) for 30 s. Detritylation was performed with one-part 800 mM 3,5-lutidine trifluoroacetate in 3:2 (v/v) DCM/ TFE + 2% (v/v) ethanol, and one-part DCM for 114 s at the same flow rate. After an 18-s DCM wash, neutralization was performed with one-part 10% NEM in 1:1 DCM/isopropanol and one-part DCM for 12 s. The resin was then washed with DCM for 24 s and DMI for 30 s, each at 5 mL/min (2.5 mL/min per pump). Coupling solution composed of one-part 0.4 M morpholino subunit in DMI and one-part 0.8 M DIEA with 0.4 M LiBr in DMI was delivered at 5 mL/min (2.5 mL/ min per pump) for 12 s (0.079 mmol monomer). DMI was delivered at 5 mL/min (2.5 mL/min per pump) until the monomer solution arrived at the reactor (12 strokes total). DMI was delivered at 0.1 mL/min for 432 s. This protocol was repeated for each residue until synthesis was complete. A final deprotection was carried out with the same detritylation reagent and conditions. The resin was removed from the reactor, washed five times with DCM in a fritted syringe (Torviq), dried under vacuum, and cleaved (see Supplementary Information, section 4 method 2). The crude products were captured on a polystyrene reverse-phase resin, and eluted with 50% aqueous acetonitrile into a preweighed 10 mL conical centrifuge tube. The sample was lyophilized to afford the crude PMO as a white powder suitable for LC–MS analysis and preparative cation exchange purification. All LC–MS data were collected and analyzed using Agilent MassHunter, and all MALDI data were collected using Bruker FlexControl, and analyzed using Bruker FlexAnalysis.

**SARS-CoV-2 inhibition assays**. All live SARS-COV-2 viral work was performed at the Iowa State University College of Veterinary Medicine's BSL-3 facility under approval by the institutional biosafety committee. SARS-CoV-2 (Seattle Strain) was obtained from Beiresources (ATCC) and expanded in Vero-E6 cells (ATCC), quantified by qRT-PCR (IDT) and frozen at −80 °C in 1 mL aliquots. Vero cells were then plated for 90% confluency in 96-well flat bottom plates a day before use. Each oligonucleotide used in this study was diluted in increasing concentrations as indicated (0, 0.01, 0.1, 1, and 10 μM) in DMEM containing 10% FBS and 1% Pen/ Strep (Gibco). These PMOs were added to some wells (duplicates to quadruplicates) containing cells 3 h prior to infection. Virus was then added at a 0.1 MOI and allowed to absorb for 2 h. Some wells had EK1 (0–10 μM, similar range as PMO) or hygromycin B (0, 10, 20, and 30 μM) instead of PMOs added to the wells during the viral incubation in varying concentrations as indicated. After the 2 h infection, cells were then rinsed of unattached virus and media replaced with fresh DMEM containing 10% FBS, 1% Pen/Strep, and similar PMO concentrations or EK1 or hygromycin B concentrations as initial conditions. Supernatants were harvested 3 days post infection and mixed 1:1 with Trizol reagent (Invitrogen), and the RNA was isolated according to the manufacturer protocols. Viral RNA was amplified on a Quant Studio 3 (Applied Biosystems) using a probe-based LUNA qRT-PCR detection system (NEB Bio) and commercial SARS-CoV-2 detection primers (IDT), according to manufacturer directions.

**Reporting summary**. Further information on research design is available in the Nature Research Reporting Summary linked to this article.

## Data availability

All the data generated or analyzed during this study are included in this published article (and in the Supplementary Information). Further details are available from the corresponding authors upon request. Source data are provided with this paper.

## Code availability

The Python code for automated operation of the flow synthesis instrument is available in a GitHub repository (https://doi.org/10.5281/zenodo.3774509).

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

## Acknowledgements

Sarepta Therapeutics is gratefully acknowledged for providing financial support for this work and the reagents for PMO production. A.J.C. is supported by the MIT School of Science Fellowship in Cancer Research. We thank Adam Chevalier and Jason Gatlin at Sarepta Therapeutics for helpful suggestions on optimizing the PMO synthesis cycle in flow. N.L.B.P. is grateful to the Radcliffe Institute for Advanced Studies for the Edward, Frances, and Shirley B. Daniels Fellow position and to Benjamin Lee (both at Harvard University) for fostering the initial software development. We thank Andrew Wilson at Detailed Dynamic for his help in the design and production of the automated instrument.

## Author contributions

C.L. and A.J.C. built the updated PMO synthesizer, improved the Python program, optimized fast-flow synthesis conditions, synthesized, and purified the PMO sequences. M.D.S. and K.A.T. performed PMO stability studies. A.J.M. and N.L.B.P. designed the open-source synthesis software, Mechwolf. A.J.M. helped build the PMO synthesizer. K.-S.P. performed the SARS-CoV-2 viral infections. D.V. designed and supervised the SARS-CoV-2 inhibition assays, and performed the viral detections. G.Z. synthesized EK1, and helped plan the SARS-CoV-2 inhibition assays. N.H. and C.K.S. helped update the PMO synthesizer. M.Z., H.Z., and G.K.H. provided input on optimization of the PMO synthesis cycle. C.L., A.J.C., M.J.S., K.A.T., and B.L.P. conceptualized the research and designed the experiments. C.L., A.J.C., A.L., M.D.S., and B.L.P. wrote the manuscript with input from all coauthors.

## Competing interests

B.L.P. is a co-founder of Amide Technologies and Resolute Bio. Both companies focus on the development of protein and peptide therapeutics. An international patent application covering part of this work has been filed by MIT and Sarepta Therapeutics (Int. Pat. Appl. WO2019060862A1). M.Z. and H.Z. are employees of Sarepta Therapeutics. K.A.T. is an employee of Amide Technologies. The authors declare no other competing interests.

## Additional information

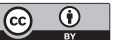

