## [Peer Review File · Nature Communications]

REVIEWER COMMENTS

Reviewer #1 (Remarks to the Author):

In this manuscript under consideration, Pentelute and coauthors report the synthesis of Phosphorodiamidate Morpholino Oligomers (PMO) using Automated Fast-Flow synthesizer. PMOs are unique class of antisense reagents, routinely used as a gene silencing reagent. It has drawn so much attention recently because PMO –based drugs are used for DMD treatment and are the only oligo-based drugs, approved by FDA. Hence its therapeutic application has made them an attractive reagent for other genetic disorder diseases and viruses.

In order to target a wide range of diseases for screening purposes, efficient synthesis of PMO is necessary like parallel DNA synthesis.

It can be only possible if the synthesis is performed by automated machine or oligo synthesizer. Author has disclosed this work in ChemRxiv 2020 earlier and now the work is under consideration for the publication in the present journal.

PMO is normally synthesized by chloroiphosphoramidate chemistry. The coupling between activated reagents with free "N" of morpholino is very slow due to the poor reactivity of pentavalent phosphorous. Additionally, presence of "NMe₂" group at the "P" center made the reagent less reactive towards "N" of morpholino ring. To improve the reactivity, LiBr has been used, however, it did not become a significant improvement so that the method can be transferred to automated synthesizer. Authors have explored the all possible conditions to improve the coupling efficiency of activated chlorophosphoramidate monomers in presence of LiBr and found 90°C is the suitable temperature for efficient coupling. A balance was strictly required between the stability of monomers and coupling efficiency because chlorophosphoramidate monomers are not stable in organic solvent particularly in presence of base at high temperature. After rigorous trials, authors have found a suitable conditions to achieve the synthesis of PMO with a high coupling efficiency within 12 min in a specially designing of automated fast flow machine. It is a significant achievement in PMO synthesis. A suitable reagent 3,5-Lut.TFA mixture has been developed for deblocking of Tr-group at 90°C so that minimization of depurination has been achieved. 18-mer sequence of PMO has been synthesized within 3.5 hrs. The synthesized PMOs have been validated in the biological assay against SARS-CoV-2 where dose-dependent inhibition was observed. It is an interesting finding in the context of the development of antiviral therapy for COVID-19.

Though, Tr-monitor is the bench mark for DNA synthesis however, this could not be done in the present case, because of using the 3,5-Lut.TFA mixture where Tr-quencher was present. Authors could monitor the progress of synthesis by Tr-assay with 3% TCA in DCM though it gives slow deprotection of Tr along with the formation of some impurities. Using of DCM at high temperature could be another issue. However, Pentelute and coauthors have executed successfully the synthesis of PMO within 3.5 hrs by automated machine starting from the design of machine using the flow chemistry technique, development of software to run the machine and chemistry. This report is very well written and conveys the information well. All compounds are well characterized, and the experimental work is described clearly. Thus, I recommend the publication of this work with the following minor corrections.

1. Authors have mentioned the pure yield of PMO. 12 mg resin bound crude PMO gives 0.5 mg pure PMO with >90% purity.

It could be better if they mention the yield with respect to the loading monomer like 12 mg (4.9 μmol , 0.39-0.43 mmol/g) resins gives 0.5 mg PMO.

2. In antiviral assay, it should be clear to understand that how much PMOs were used for inhibition because from the following sentences, it means the two times PMOs (0-10 μM) were added.

Lines 321 to 324: Vero cells were then plated for 90% confluency in 96 well flat bottom plates 321 a day before use with each oligonucleotide used in this study diluted in varying concentrations as indicated (0-10 μM) in DMEM 10% FBS 1% Pen/Strep (Gibco). These PMOs were added to some wells (duplicates to quadruplicates) containing cells after they are attached.

Lines 327 to 329: Cells were then rinsed and media replaced with fresh DMEM 10% FBS 1% Pen/Strep containing similar PMO concentrations or EK1 or hygromycin B. Supernatants were harvested 3 days post-infection, mixed 1:1 with Trizol reagent.

Reviewer #2 (Remarks to the Author):

In this manuscript, the authors outlined full-automatic synthesis of antisense phosphorodiamidate morpholino oligomers (PMOs) using flow reactors. Because their design is informational in nature, antisense therapeutics can target virtually any genetic disease by binding to mRNA via complementary base-pairing interactions. PMOs are the most tested as antiviral agents among antisense compound types because of their rigid structure. However, tedious synthesis protocols for PMOs limit their production and development.

The authors overcame the limitation using solid-phase oligomerization of PMO monomers performed by automated flow reactors. Their optimized synthesis platform reduced PMO elongation reaction times by about 20-fold compared to the previous report. This acceleration totally allows development of PMOs, and the authors demonstrated this technique synthesizing three new candidates therapeutic PMO for Duchenne muscular dystrophy (DMD), and PMO that targets the genomic mRNA of SARS-CoV-2. Given the urgent circumstances around the global COVID-19 pandemic, the fast PMO synthesis must be important and be of wide interest.

Although I really agree the contribution of this report developing anti-SARS-CoV-2 PMO, I am sorry to say that this report seems moderate significance in organic synthesis field. Solid-phase synthesis for biological oligomers has been widely used to develop lots of oligomeric peptides, sugars, and so on (as the authors referred in ref 29 and 30). The PMOs-synthesis procedure developed by the authors is almost same to those previous ones where protected monomers are reacted to an oligomer-end followed by a deprotection. Since the solid-phase synthesis for PMO has been already reported (for example, doi.org/10.26434/chemrxiv.12765563.v1), this present synthetic methodology seems to be an extent of those previous reports. The machine setup seems almost similar to the authors previous report (ref 29), and the idea for automated synthesis using flow reactors had been already described in ref 29. Because the synthetic tactic seems similar, the authors, I guess, should mention the novelty of this present procedure, otherwise I am afraid that this report resulted in a mere extension of the substrate scope.

The manuscript is well written and replete with an extensive bibliography, the experimental results are of great importance to the development of PMOs, and the supporting information is also of a high quality. Because of the importance of antisense POMs, publication should be recommended although I guess the synthetic novelty is moderate.

AUTHOR'S RESPONSE TO REVIEWERS

Reviewer #1 (Remarks to the Author):

Comment:

In this manuscript under consideration, Pentelute and coauthors report the synthesis of Phosphorodiamidate Morpholino Oligomers (PMO) using Automated Fast-Flow synthesizer. PMOs are unique class of antisense reagents, routinely used as a gene silencing reagent. It has drawn so much attention recently because PMO –based drugs are used for DMD treatment and are the only oligo-based drugs, approved by FDA. Hence its therapeutic application has made them an attractive reagent for other genetic disorder diseases and viruses.

In order to target a wide range of diseases for screening purposes, efficient synthesis of PMO is necessary like parallel DNA synthesis.

It can be only possible if the synthesis is performed by automated machine or oligo synthesizer. Author has disclosed this work in ChemRxiv 2020 earlier and now the work is under consideration for the publication in the present journal.

PMO is normally synthesized by chlorophosphoramidate chemistry. The coupling between activated reagents with free “N” of morpholino is very slow due to the poor reactivity of pentavalent phosphorous. Additionally, presence of “NMe₂” group at the “P” center made the reagent less reactive towards “N” of morpholino ring. To improve the reactivity, LiBr has been used, however, it did not become a significant improvement so that the method can be transferred to automated synthesizer. Authors have explored the all possible conditions to improve the coupling efficiency of activated chlorophosphoramidate monomers in presence of LiBr and found 90oC is the suitable temperature for efficient coupling. A balance was strictly required between the stability of monomers and coupling efficiency because chlorophosphoramidate monomers are not stable in organic solvent particularly in presence of base at high temperature. After rigorous trials, authors have found a suitable conditions to achieve the synthesis of PMO with a high coupling efficiency within 12 min in a specially designing of automated fast flow machine. It is a significant achievement in PMO synthesis. A suitable reagent 3,5-Lut.TFA mixture has been developed for deblocking of Tr-group at 90oC so that minimization of depurination has been achieved. 18-mer sequence of PMO has been synthesized within 3.5 hrs. The synthesized PMOs have been validated in the biological assay against SARS-CoV-2 where dose-dependent inhibition was observed. It is an interesting finding in the context of the development of antiviral therapy for COVID-19. Though, Tr-monitor is the bench mark for DNA synthesis however, this could not be done in the present case, because of

Bradley L. Pentelute
Associate Professor
Phone: (617) 324-0180
Email: blp@mit.edu

Massachusetts Institute of Technology
Department of Chemistry, Room 18-596 77
Massachusetts Avenue
Cambridge, MA 02139-4307

using the 3,5-Lut.TFA mixture where Tr-quencher was present. Authors could monitor the progress of synthesis by Tr-assay with 3% TCA in DCM though it gives slow deprotection of Tr along with the formation of some impurities. Using of DCM at high temperature could be another issue. However, Pentelute and coauthors have executed successfully the synthesis of PMO within 3.5 hrs by automated machine starting from the design of machine using the flow chemistry technique, development of software to run the machine and chemistry. This report is very well written and conveys the information well. All compounds are well characterized, and the experimental work is described clearly. Thus, I recommend the publication of this work with the following minor corrections.

Our response:

We thank the reviewer for their evaluation of our work and the final recommendation for publication.

Comment:

1. Authors have mentioned the pure yield of PMO. 12 mg resin bound crude PMO gives 0.5 mg pure PMO with >90% purity. It could be better if they mention the yield with respect to the loading monomer like 12 mg (4.9 μmol , 0.39-0.43 mmol/g) resins gives 0.5 mg PMO.

Our response:

We adjusted how we report synthesis yields in line with the reviewer's recommendation to add clarity. We revised the manuscript as follows:

New Lines 213-215:

"From each synthesis 1 mg of PMO with more than 85% purity were obtained (10.1%, 10.1%, and 10.2% yields with respect to resin loading, 10 mg of 0.39-0.43 mmol/g) (Fig. 4c and 4e)."

New Lines 241-244:

"Synthesis and subsequent purification of one third of the crude sample yielded 0.5 mg of PMO with >90% purity (extrapolated yield of 5.5% with respect to resin loading, 10 mg of 0.39-0.43 mmol/g) (Fig. 6b)."

Comment:

2. In antiviral assay, it should be clear to understand that how much PMOs were used for inhibition because from the following sentences, it means the two times PMOs (0-10 μM) were added.

Lines 321 to 324: Vero cells were then plated for 90% confluency in 96 well flat bottom plates 321 a day before use with each oligonucleotide used in this study diluted in varying concentrations as indicated (0-10 μM) in DMEM 10% FBS 1% Pen/Strep (Gibco). These PMOs were added to some wells (duplicates to quadruplicates) containing cells after they are attached.

Lines 327 to 329: Cells were then rinsed and media replaced with fresh DMEM 10% FBS 1% Pen/Strep containing similar PMO concentrations or EK1 or hygromycin B. Supernatants were harvested 3 days post-infection, mixed 1:1 with Trizol reagent.

Our response:

Bradley L. Pentelute
Associate Professor
Phone: (617) 324-0180
Email: blp@mit.edu

Massachusetts Institute of Technology
Department of Chemistry, Room 18-596 77
Massachusetts Avenue
Cambridge, MA 02139-4307

We thank the reviewer for pointing out the lack of clarity in the description of the infectivity assay and have revised our methods section to accurately describe the treatment concentrations. Specifically, we clarified how much treatment PMO was added in each condition. Cells were incubated with PMO both pre and post infection. This amounts to treating the cells two times as the reviewer points out. This practice is required because the culture media must be replaced after viral infection. In addition, this practice of pre- and post-treatment is consistent with earlier reports of this assay format as shown in reference 15 (Neuman, B. W. *et al.* Inhibition, escape, and attenuated growth of severe acute respiratory syndrome coronavirus treated with antisense morpholino oligomers. *J. Virol.* **79**, 9665-9676 (2005)). The revised methods section now reads:

“Each oligonucleotide used in this study was diluted in increasing concentrations as indicated (0, 0.01, 0.1, 1, and 10 μ M) in DMEM containing 10% FBS and 1% Pen/Strep (Gibco). These PMOs were added to some wells (duplicates to quadruplicates) containing cells 3 hours prior to infection. Virus was then added at a 0.1 multiplicity of infection (MOI) and allowed to absorb for 2 hours. Some wells had EK1 (0-10 μ M, similar range as PMO) or hygromycin B (0, 10, 20, and 30 μ M) instead of PMOs added to the wells during the viral incubation in varying concentrations as indicated. After the 2 hour infection, cells were then rinsed of unattached virus and media replaced with fresh DMEM containing 10% FBS, 1% Pen/Strep, and similar PMO concentrations or EK1 or hygromycin B concentrations as initial conditions.”

Reviewer #2 (Remarks to the Author):

Comment:

In this manuscript, the authors outlined full-automatic synthesis of antisense phosphorodiamidate morpholino oligomers (PMOs) using flow reactors. Because their design is informational in nature, antisense therapeutics can target virtually any genetic disease by binding to mRNA via complementary base-pairing interactions. PMOs are the most tested as antiviral agents among antisense compound types because of their rigid structure. However, tedious synthesis protocols for PMOs limit their production and development. The authors overcame the limitation using solid-phase oligomerization of PMO monomers performed by automated flow reactors. Their optimized synthesis platform reduced PMO elongation reaction times by about 20-fold compared to the previous report. This acceleration totally allows development of PMOs, and the authors demonstrated this technique synthesizing three new candidates therapeutic PMO for Duchenne muscular dystrophy (DMD), and PMO that targets the genomic mRNA of SARS-CoV-2. Given the urgent circumstances around the global COVID-19 pandemic, the fast PMO synthesis must be important and be of wide interest. Although I really agree the contribution of this report developing anti-SARS-CoV-2 PMO, I am sorry to say that this report seems moderate significance in organic synthesis field. Solid-phase synthesis for biological oligomers has been widely used to develop lots of oligomeric peptides, sugars, and so on (as the authors referred in ref 29 and 30). The PMOs-synthesis procedure developed by the authors is almost same to those previous ones where protected monomers are reacted to an oligomer-end followed by a deprotection. Since the solid-phase synthesis for PMO has been already reported (for example, doi.org/10.26434/chemrxiv.12765563.v1), this present synthetic methodology seems to be an extent of those previous reports. The machine setup seems almost similar to the authors previous report (ref 29), and the idea for automated synthesis using flow reactors had been already described in ref 29. Because the synthetic tactic seems similar, the authors, I guess, should mention the novelty of this present procedure, otherwise I am afraid that this report resulted in a mere extension of the substrate scope. The manuscript is well written and replete with an extensive bibliography, the experimental results are of great importance to the development of PMOs, and the supporting information is also of a high quality. Because of the importance of antisense POMs, publication should be recommended although I guess the synthetic novelty is moderate.

Bradley L. Pentelute
Associate Professor
Phone: (617) 324-0180
Email: blp@mit.edu

Massachusetts Institute of Technology
Department of Chemistry, Room 18-596 77
Massachusetts Avenue
Cambridge, MA 02139-4307

Our response:

We thank the reviewer for their thoughtful analysis of our work, and their final recommendation for publication. We appreciate that the reviewer placed the novelty of our work in context of what has already been reported in solid-phase synthesis. With respect to their comments about the novelty of solid-phase synthesis of PMO in flow specifically, we would like to point out that the reference the reviewer provides in fact cites our own work described in this manuscript as a preprint (see reference 17 in the preprint cited by the reviewer, available at: doi.org/10.26434/chemrxiv.12765563.v1). Therefore, the study cited by the reviewer is related to and inspired by our original work posted in preprint form in July 2020. As this reviewer astutely points out, solid-phase PMO synthesis has been reported before. What has not been reported before however, is a high-temperature automated PMO synthesis in flow that would enable rapid couplings. In order to achieve the synthesis at high temperature in flow, we had to adapt existing chemistry and design an entirely new instrument to carry it out, innovations that go beyond simple extensions of prior work.

As the reviewer points out, the automated synthesizer we report is of a similar design to a previously reported instrument built in our laboratory (highlighted in ref. 29). We actually highlight this fact ourselves in lines 109-110 of our manuscript. Importantly, there are a series of major changes that we go on to describe in lines 110-124 that clearly differentiate our new instrument. Amongst the most notable are a custom micro-scale reaction chamber, and an entirely different software control package. These changes are critical to enabling PMO synthesis specifically.

The revisions to our manuscript are clearly highlighted in the enclosed manuscript with tracked changes. With these changes in place, we believe our manuscript is now suitable for publication and we thank you for your consideration of our manuscript.